# RIP5 Interacts with REL1 and Negatively Regulates Drought Tolerance in Rice

**DOI:** 10.3390/cells13110887

**Published:** 2024-05-21

**Authors:** Qiuxin Zhang, Dan He, Jingjing Zhang, Hui He, Guohua Guan, Tingting Xu, Weiyan Li, Yan He, Zemin Zhang

**Affiliations:** State Key Laboratory for Conservation and Utilization of Subtropical Agro-Bioresources, Guangdong Provincial Key Laboratory of Plant Molecular Breeding, South China Agricultural University, Guangzhou 510642, China; zhangqiuxin@stu.scau.edu.cn (Q.Z.); hedan001@stu.scau.edu.cn (D.H.); zjjsacu@scau.edu.cn (J.Z.); huihcarrie@163.com (H.H.); 609994023@stu.scau.edu.cn (G.G.); 20221015021@stu.scau.edu.cn (T.X.); 20211015008@stu.scau.edu.cn (W.L.); hy9509@126.com (Y.H.)

**Keywords:** ascorbate oxidase, drought response, rice, transcriptome

## Abstract

Improving the drought resistance of rice is of great significance for expanding the planting area and improving the stable yield of rice. In our previous work, we found that *ROLLED AND ERECT LEAF1* (*REL1*) protein promoted enhanced tolerance to drought stress by eliminating reactive oxygen species (ROS) levels and triggering the abscisic acid (ABA) response. However, the mechanism through which REL1 regulates drought tolerance by removing ROS is unclear. In this study, we identified REL1 interacting protein 5 (RIP5) and found that it directly combines with REL1 in the chloroplast. We found that *RIP5* was strongly expressed in ZH11 under drought-stress conditions, and that the *rip5-ko* mutants significantly improved the tolerance of rice plants to drought, whereas overexpression of *RIP5* resulted in greater susceptibility to drought. Further investigation suggested that *RIP5* negatively regulated drought tolerance in rice by decreasing the content of ascorbic acid (AsA), thereby reducing ROS clearance. RNA sequencing showed that the knockout of *RIP5* caused differential gene expression that is chiefly associated with ascorbate and aldarate metabolism. Furthermore, multiple experimental results suggest that *REL1* is involved in regulating drought tolerance by inhibiting *RIP5*. Collectively, our findings reveal the importance of the inhibition of RIP5 by REL1 in affecting the rice’s response to drought stress. This work not only explains the drought tolerance mechanism of rice, but will also help to improve the drought tolerance of rice.

## 1. Introduction

Rice is the main food source for more than half of the world’s population. In recent years, global warming, drought, high temperatures, and other natural disasters have occurred frequently, seriously affecting the yield and quality of rice [1]. Rice, China’s most significant food crop, which is irrigated with almost 65% of the country’s agricultural water use, is a water-intensive crop [2]. China has freshwater resources per person of 2400 m^3^, making it one of the 13 countries in the world with low water availability [3]. Given the water scarcity issue that China’s rice industry is currently experiencing, it is extremely important from scientific and practical standpoints to investigate the genes that confer drought tolerance in rice, examine the mechanisms behind drought tolerance, and develop drought-tolerant rice varieties.

Plants have evolved a variety of efficient protection mechanisms in the process of adapting to drought. These drought-related regulatory mechanisms are extremely complex. There are a variety of drought-response genes involved in this regulation, which affect a variety of physiological and biochemical reactions and a series of signal transduction pathways [4]. Drought stress damages plant tissues, exacerbates the process of self-oxidation, and raises the concentration of reactive oxygen species (ROS) in plants. Plants have evolved non-enzymatic ROS-scavenging systems to maintain the homeostasis of ROS, which mainly include ascorbic acid (AsA), glutathione (GSH), mannitol, flavonoids, isoprenoids, carotenoids, tocopherol (vitamin E), ubiquinone, and plastoquinone [5]. The ascorbate-glutathione cycle (AsA-GSH) is an important antioxidant pathway for ROS clearance, and ascorbate oxidase (AAO) is a group of species found in plants and some fungi that belongs to the blue copper oxidase family and is also important in the AsA-GSH circulatory system [6]. Ascorbic acid (AsA) is one of the most important antioxidant substances in plant cells, and it can clear ROS through a reduction reaction. Increasing the AsA content in plants could enhance their stress resistance [6]. Ascorbate oxidase breaks down AsA to generate dehydroascorbate (DHA). Some studies have shown that increased expression of genes associated with ROS clearance can increase tolerance to drought stress. For example, overexpression of *SNAC3* regulates ROS homeostasis by regulating the expression of ROS clearance related genes, increasing drought tolerance in rice. In addition, *OsDIS1* is a negative regulator of drought-stress signaling through the suppression of the reactive oxygen species-scavenging pathway and stomatal opening in rice [7].

AsA is one of the most universally water-soluble antioxidant molecules. An enzyme named AAO, belonging to the blue copper oxidase enzyme family, oxidizes AsA in plant cells. This enzyme is generally present in the apoplastic space of plant cells, where it functions as a terminal oxidase in conjunction with other oxidation-reduction reactions. It can catalyze the oxidation of AsA to produce water and dehydroascorbic acid [8,9]. Genome-wide analysis of the AAO family revealed different gene members in various plant species. For example, seven and five AAO genes were identified in Arabidopsis and rice, respectively. Ascorbate oxidase genes were regulated by developmental cues and various stress conditions [10]. Overexpression of AAO alters ascorbate and glutathione redox states and enhances sensitivity to ozone in tobacco [11]. Tobacco plants inserted with antisense Arabidopsis AAO displayed low ascorbate oxidase activity and increased tolerance for salt stress during the vegetative and reproductive stages [12]. In rice, *OsAAO2 (RIP5)* was one of the most stress-responsive genes in shoot tissues and *OsAAO3* and *OsAAO4* were highly responsive when induced under salinity and drought stress [10]. A recent study revealed that AAO activates systemic defense against the root-knot nematode Meloidogyne graminicola through a process dependent on both jasmonic acid (JA) and ethylene (ET) pathways in rice [13]. Furthermore, *OsAAO2 (RIP5)* is reported to enhance basal ROS-mediated resistance to RSV by oxidizing AsA, the primary ROS scavenger [14]. Although the roles of a few AAOs have been investigated in rice, knowledge of the underlying molecular mechanisms of AAO in regulating abiotic stress remains largely limited.

Previously, we identified a rice T-DNA insertion gain-of-function mutation named *ROLLED AND ERECT LEAF1* (*REL1*) that was extensively involved in leaf morphology and abiotic-stress resistance, which had significantly increased expression under conditions of drought stress [14]. *REL1* encodes an unknown functional domain that improves tolerance to drought stress by reducing ROS levels and inducing abscisic acid (ABA) responses [15,16,17]. In this study, we determined that REL1 physically interacts with RIP5 in the chloroplasts. We constructed three types of transgenic plants, including *rip5-ko*, *RIP5-OE*, and *rel1-D/rip5-ko* mutant lines, for the experiments used in this study. Our results indicate that *RIP5* is inhibited by *REL1* to negatively regulate rice drought tolerance through antioxidant pathways.

## 2. Materials and Methods

### 2.1. Obtainment of the Mutants

ZH11 used in this paper is a kind of japonica conventional rice, which was selected and bred by the Institute of Crop Science at the Chinese Academy of Agricultural Sciences. The *rel1-D* strain, a rice strain with a T-DNA-insertion gain-of-function mutation, has been obtained previously by our research group [15]. The *rip5-ko* and *rel1-D/rip5-ko* mutant strains were generated by knocking out *RIP5* using CRISPR/Cas9 technology [18]. The sgRNA target sites (GGATAAACGGCAGGTTCCCGGGG) were designed in order to construct the *RIP5*-Cas9 vector, which was transformed into ZH11 and *rel1-D* using an Agrobacterium-mediated method by the Wuhan Boyuan Company. The genotypes of CRISPR/Cas9 plants were analyzed by PCR and by the direct sequencing of the amplification products. The full-length coding sequence of *RIP5* was cloned into the pCXSN vector using homologous recombination to generate the pCXSN-*RIP5* vector, for which expression was driven by the 35S promoter. *RIP5-OE* was generated by Agrobacterium-mediated transformation and then transformed into ZH11 by the Wuhan Boyuan Company. After propagation to the T_2_ generation and after identification, the stress experiment was carried out. All primers used in this study are described in Appendix A.

### 2.2. Plant Materials and Growth Conditions

*Oryza sativa* L. *japonica* ‘Zhonghua 11’ (ZH11) was used as the wild-type strain for various experiments. Rice plant strains, including ZH11, *rip5-ko*, *RIP5-OE*, *rel1-D,* and *rel1-D/rip5-ko* used in this study were grown and propagated in the net room of the farm of the South China Agriculture University at Guangzhou, China. Phenotypic characteristics were measured and recorded during the natural growing seasons. For the experiments examining drought-stress treatments, rice plants, including ZH11, *rip5-ko*, *RIP5-OE*, *rel1-D* and *rel1-D*/*rip5-ko* strains, were grown in a greenhouse room under 14 h light/10 h dark cycles at 28 °C.

### 2.3. Yeast Two-Hybrid Assays (Y2H) and BIMO-Lecular Fluorescence Complementation (BiFC)

Yeast two-hybrid assays were performed according to the method described by Peng et al. [19]. The full-length *RIP5* gene was amplified from ZH11 and cloned into the pGADT7 vector (Takara, Osaka, Japan), and the full-length *REL1* gene was inserted into the pGBKT7 vector (Takara, Osaka, Japan). The vectors were co-transformed into yeast strain AH109. The primers used are listed in Appendix A.

For BiFC, the *RIP5* coding sequence without a stop codon was amplified and inserted into the pSPYCE vector to produce fusions with the C-terminal fragment of yellow fluorescent protein (YFP) [19]. The full-length *REL1* gene was cloned into pSPYNE to produce fusions with the N-terminal fragment of YFP. Rice protoplasts were extracted and transformed using a modified version of the method by Chen et al. [20]. The YFP and chloroplast auto-fluorescence signals (Chlorophyll) were observed and photographed using a confocal microscope (Leica, Wetzlar, Germany). The primers used are listed in Appendix A.

### 2.4. Bioinformatics Analysis

The sequence of *REL1 interacting protein 5* (*RIP5*) obtained from the Y2H screening experiment was aligned with the genome of Nipponbare in NCBI BLAST (https://blast.ncbi.nlm.nih.gov/Blast.cgi, accessed on 10 May 2024). DNAMAN was used to align the sequence of *OsAAO2* and *RIP5*_partial_region obtained from the Y2H-screening experiment. The gene and amino acid sequences of *RIP5* were obtained from UNIPORT (https://www.uniprot.org, accessed on 15 April 2024), and the protein domain of *RIP5* was predicted from SMART (https://smart.embl.de/, accessed on 15 April 2024). The Hidden Markov model (HMM) corresponding to the domain was obtained from Interpro (https://www.ebi.ac.uk/interpro/, accessed on 15 April 2024), and the homologous genes were matched from the rice whole genome annotation file by Hmmer 3.0. Mafft was used to conduct amino acid sequence comparison for homologous genes using a “G-INS-I (accurate)” strategy. According to the sequence comparison results, FastTree was used to construct phylogenetic tree based on the maximum likelihood method. Expasy (https://www.expasy.org/, accessed on 15 April 2024) was used to obtain the physicochemical properties of homologous genes, and Plant-PLoc (https://journals.plos.org/, accessed on 15 April 2024) was used to predict the subcellular localization of homologous genes. The genes homologous to *RIP5* in arabidopsis, soybean, sorghum, maize, wheat, rape, and tomato were obtained by using the submodule BioMart of Ensembl plants (https://plants.ensembl.org, accessed on 15 April 2024), and their full-length amino acid sequences were compared to analyze their conserved properties.

### 2.5. Subcellular Localization of RIP5

The coding region of *RIP5* was amplified from ZH11 and inserted into the pRTVnGFP vector. Protoplasts were isolated using a protocol modified from that of Chen et al. [20]. The green fluorescent protein (GFP) and the Chlorophyll fluorescence signal of the chloroplasts were observed and photographed using a confocal microscope (Leica SP8, Germany).

### 2.6. Total RNA Isolation and Quantitative Real-Time PCR

Total rice RNA was extracted for three biological replicates using the RNA Extraction Kit (TRIgol reagent, BEIJING DINGGUO, Beijing, China) according to the manufacturer’s protocol. First-strand cDNA (Complementary DeoxyriboNucleic Acid) was synthesized from 2 μg of total RNA using 5 × All-In-One RT Master Mix with the AccuRT Genomic DNA (DeoxyriboNucleic Acid) Removal Kit (Applied Biological Materials Inc., Vancouver, BC, Canada). Quantitative real-time PCR (qRT-PCR) was performed using Applied Biosystems™ SYBR™ Green PCR Mix on the Bio-Rad CFX96 system (Bio-Rad, Hercules, CA, USA). The rice *OsActin1* gene was used as an internal control, and the relative expression levels of the target genes were analyzed using the 2^−ΔΔCT^ method. All primers used in qRT-PCR are listed in Appendix A.

### 2.7. Expression Profile Analysis of RIP5 and AAO families

For the expression analysis of *RIP5*, total RNA was extracted for three biological replicates from various tissues of ZH11 at different developmental stages. The tissues included those from the root and leaf at seedling stage; from the root, leaf, stem, and sheath at the tillering stage; from young panicles at the booting stage; and from flag leaf, stem, and sheath at the filling stage. These tissues were then used for RNA extraction, followed by qRT-PCR analysis. In addition, two-week-old ZH11 seedlings were exposed to PEG 6000 (20%, *w*/*v*) at 0 h, 1 h, 3 h, 6 h or 9 h after treatment and used for RNA extraction for qRT-PCR analysis to assess the AAO family-expression pattern during the drought-stress process.

### 2.8. Drought-Stress Treatment

For drought-stress testing, wild type ZH11, *rip5-ko*, *RIP5-OE*, *rel1-D* and *rel1-D*/*rip5-ko* transgenic lines (more than 20 plants in each line) were grown in soil for 1 month and then exposed to simulated drought stress by halting irrigation for 14 d, followed by rewatering for 7 d. After recovery for 7 d, plants that had green leaves were considered survivors. These data were used to calculate the overall survival rates. The surviving seedlings were counted and photographed, with three biological replicates assessed per line.

### 2.9. Ascorbate Oxidase (AAO) Activity Assay and Ascorbic Acid (AsA) Measurement

The AAO activity was measured with three biological replicates (*n* = 20) using an AAO activity test kit (Nanjing jiancheng, Nanjing, China) according to the manufacturer’s instructions. The protein concentration was measured using the Bradford method, and the activity was calculated based on the protein concentration. Total AsA was extracted from three biological replicates (*n* = 20) with 1 mL trichloroacetic acid (6% TCA, *w*/*v*) by centrifugation for 20 min at 4 °C and 15,000× *g*. The absorbance of the red complex compound was recorded at 550 nm after binding with bipyridine by AsA to oxidize Fe^3+^. AsA content was calculated using an AsA standard curve.

### 2.10. RNA-Sequencing and Data Analysis

Total RNA was extracted for three biological replicates from leaves under normal and PEG-treatment conditions (20% PEG 6000, *w*/*v*). Three biological replicates were prepared for each rice plant. RNA-seq libraries were constructed with the TruSeq RNA Kit (Illumina, San Diego, CA, USA) and sequenced on a HiSeq X Ten sequencer by Novogene Bioinformatics Technology Company (Beijing, China). After filtering adapters and low-quality reads, the paired-end reads were mapped against the Nipponbare reference genome (IRGSP-1.0). Cufflinks were used to estimate gene expression levels via fragments per kilobase of transcript per million fragments mapped (FPKM). The differentially expressed genes (DEGs) were determined using the criteria of |log2(fold change)| ≥ 1 and padj < 0.05. Representative DEGs were confirmed by qRT-PCR. All primers used in qRT-PCR are listed in Appendix A.

### 2.11. Data Analysis

All data are presented as the mean ± SD. The Shapiro–Wilk test was used to assess the normal distribution of all data. The analyses comparing two groups of data were carried out by Student’s *t*-test, and LSD and Duncan’s tests were used for multiple-group tests.

## 3. Results

### 3.1. REL1 Physically Interacts with RIP5

The *rel1-D* strain, a rice strain with a T-DNA-insertion gain-of-function mutation, regulates leaf rolling and erectness in rice and displays enhanced tolerance to drought stress by reducing ROS levels and triggering an ABA response. To further explore the mechanisms of drought stress in rice, we used REL1 as a bait to screen a cDNA library and identify REL1-interacting proteins (RIPs). We screened an interacting protein of REL1 and named it REL1-interacting protein 5 (RIP5).

We used the cDNA of RIP5 with the Y2H to confirm the interaction between REL1 and RIP5. The results showed that pGADT7-RIP5 and pGBKT7-REL1 co-transformed into AH109 and grew on QDO/-Leu/-Trp/-His/-Ade medium, which was consistent with the pGADT7-T and pGBKT7-53, the positive controls. (Figure 1A). Moreover, BiFC analysis was also used to confirm an in vivo interaction between REL1 and RIP5. The results showed that fluorescence only occurred in the chloroplast of rice protoplast cells when pSPYNE-REL1 and pSPYCE-RIP5 were co-expressed, whereas there was no fluorescence in negative controls, including pSPYNE with pSPYCE, pSPYNE with pSPYCE-RIP5, and pSPYNE-REL1 with pSPYCE. (Figure 1B). These results imply that REL1 and RIP5 interact in chloroplasts to contribute to the drought resistance of rice.

### 3.2. Phylogenetic Analysis of RIP5 and Expression Patterns of OsAAOs Family to Drought Stress

We found that the sequence of *REL1 interacting protein 5* (*RIP5*) obtained from the Y2H screening experiment was 100 % consistent with only the Chr 6 of Nipponbare (Chr 6: 21954299-21954934) through NCBI BLAST (Appendix A). This physical location exists at *OsAAO2* (Chr 6: 21951201-21955132) referring to the genome annotation file of Nipponbare. Furthermore, we found that the similarity between *RIP5* and *OsAAO2* partial sequence was 100 % (Appendix A). These results indicate that *RIP5* is *OsAAO2*. 

To further investigate the function of *RIP5* (*LOC_Os06g37150/OsAAO2*), we further analyzed the structure; physiological and biochemical properties; and gene family of RIP5, and we predicted the protein structure of the gene family. The open reading frame of *RIP5* has a total length of 5472 bp, and the CDS (Coding sequence) has a total length of 1902 bp, encoding 633 amino acids. RIP5 consists of four exons and three introns and contains a signal peptide domain and three Cu-oxidase domains predicted by SMART, including Cu-oxidase (PF00394), Cu-oxidase_2 (PF07731), and Cu-oxidase_3 (PF07732) (Appendix A). RIP5 is unstable, with a predicted molecular mass of 69.12 kDa, pI of 5.90, and instability index of 43.86 (Appendix A). However, the role of *RIP5* in regulating drought tolerance remains obscure.

We aligned the sequence and found that there are six OsAAOs family members in rice, including Os06t0567200-01 (OsAAO1), Os06t0567900-01 (RIP5/OsAAO2), Os07t0119400-01 (OsAAO3), Os09t0365900-01 (OsAAO4), Os09t0507300-01 (OsAAO5), and Os01t0816700-01 (OsAAO6) (Figure 2A). We then identified the homologous genes of *RIP5* in arabidopsis, soybean, sorghum, maize, wheat, rape, and tomato by using the submodule BioMart of Ensembl plants, and we compared their full-length amino acid sequences to analyze their conserved properties (Appendix A). The alignment demonstrated that regions containing a Cu-oxidase domain (36–153 aa), a Cu-oxidase_2 domain (168–361 aa), and a Cu-oxidase_3 domain (449–595 aa) were highly conserved among all species (Appendix A).

In order to clarify the response mechanism of OsAAOs family members to drought stress in rice, we examined the expression pattern of *OsAAOs* under PEG treatment. The results showed that *OsAAOs* had different expression patterns in drought stress. The expression trend of *RIP5 (OsAAO2)* and *OsAAO5* was consistent, increasing in the first 3 h and then going down, suggesting that *RIP5* is induced by drought and may participate in drought stress (Figure 2C,F). In addition, the expression trend of *OsAAO1* was consistent with that of *OsAAO6*, whereas the expression patterns of *OsAAO3* and *OsAAO4* were unique (Figure 2B,D,E,G). The relative expression of *OsAAO1* and *OsAAO6* decreased at first, then increased, and then finally decreased, reaching the maximum at 3 h after drought-stress treatment commenced (Figure 2B,G). Taken together, the expression of these *OsAAOs* in drought-stress tolerance suggests that they may play potential roles at different times or in different degrees of drought treatment.

### 3.3. The Expression Characteristics Analysis and Subcellular Localization of RIP5

We analyzed the expression characteristics of *RIP5*. The results show that *RIP5* was highly expressed in the leaves, stems, and sheaths at the tillering stage (Figure 3A). The BiFC showed REL1 and RIP5 interacting in chloroplasts. In order to further confirm whether RIP5 is localized in chloroplasts, the ORF of the *RIP5* gene was fused to the 5’ end of the green fluorescent protein (GFP) under a ubiquitous promoter and transiently expressed in rice-leaf protoplasts. The fluorescence signal of RIP5 was localized within chloroplasts, whereas green fluorescence was visible throughout the cytoplasm of rice protoplasts for the control plasmid, indicating that RIP5 was localized in the chloroplasts (Figure 3B).

### 3.4. RIP5 Negatively Regulates Drought-Stress Tolerance in Rice

To assess the biological function of RIP5 in rice, we generated *rip5-ko* mutants in the ZH11 cultivar background using CRISPR/Cas9 technology. The target site was located in the first exon of *RIP5* (Figure 4A). Two homozygous lines were selected for further analysis, including *rip5-ko#6* with a 1 bp deletion (-C) and *rip5-ko#12* with a 2 bp deletion (-TC) (Figure 4A,B). Furthermore, we also identified that the nucleotide sequences of *RIP5* CDS from two *rip5-ko* mutants had frame shift mutations, and their nucleotide sequences were only 69 aa and 222 aa, respectively, which was caused by premature termination. We applied drought-stress treatments to one-month seedlings of ZH11 and *rip5-ko* rice strains. After 2 weeks of drought stress followed by rewatering for a week, phenotypic observation showed that the survival rate of ZH11 was 19.56%, the survival rate of *rip5-ko#6* was 29.60%, and the survival rate of *rip5-ko#12* was 26.83%. Both *rip5-ko#6* and *rip5-ko#12* strains showed significantly higher survival rates compared to ZH11 (Figure 4C,D). These results indicate that knocking out *RIP5* enhances drought-stress tolerance in rice.

Additionally, we produced *RIP5*-overexpressing lines in the ZH11 cultivar background and obtained two independent transformants (*RIP5-OE#10*, *#18*) in which the relative expression of *RIP5* was significantly increased (Figure 5B). The same drought-stress treatment was applied to the *RIP5-OE* lines, and an obvious difference was detected between *RIP5-OE* and ZH11 (Figure 5A). The survival rate of ZH11 was 19.56%, survival rate of *RIP5-OE#10* was 4.10%, and survival rate of *RIP5-OE#18* was 3.70%. Both *RIP5-OE#10* and *RIP5-OE#18* displayed significantly decreased survival rates compared to ZH11 (Figure 5C). Taken together, these results show that *RIP5* negatively regulates drought-stress tolerance in rice.

### 3.5. RIP5 Negatively Regulates Drought Response by Affecting the Content of AsA

To further investigate the involvement of RIP5 in AAO activity under drought-stress, we measured AAO activity and AsA content in ZH11, *rip5-ko* and *RIP5-OE* lines under normal growth conditions and under drought stress. The AAO activity of ZH11 was 40.88 U/mg_prot, the AAO activity of *rip5-ko#6* was 19.86 U/mg_prot, the AAO activity of *rip5-ko#12* was 23.88 U/mg_prot, the AAO activity of *RIP5-OE#10* was 1657.40 U/mg_prot, and the AAO activity of *RIP5-OE#18* was 945.82 U/mg_prot. The results showed that the AAO activities of *rip5-ko#6* and *rip5-ko#12* were significantly decreased compared to ZH11, whereas those of *RIP5-OE#10* and *RIP5-OE#18* were significantly increased under drought-stress conditions (Figure 6A). In addition, the AsA content of ZH11 was 2.90 μg/g, the AsA content of *rip5-ko#6* was 5.61 μg/g, the AsA content of *rip5-ko#12* was 10.52 μg/g, the AsA content of *RIP5-OE#10* was 2.07 μg/g, and the AsA content of *RIP5-OE#18* was 1.99 μg/g. On the contrary, the AsA content was significantly increased in two *rip5-ko* lines compared to ZH11 under drought-stress conditions (Figure 6B). These results suggest that *RIP5* negatively regulates the drought tolerance of rice by oxidizing AsA.

### 3.6. REL1 Is Involved in Regulating Drought Tolerance by Inhibiting RIP5

To study the regulatory relationship between REL1 and RIP5, we knocked out *RIP5* in a *rel1-D* mutant background to produce a *rel1-D/rip5-ko* double mutant. We further applied drought-stress treatment to ZH11, *rip5-ko*, *RIP5-OE*, *rel1-D*, and *rel1-D/rip5-ko* lines. The results showed the following survival rates: 21.06% for ZH11, 29.56% for *rip5-ko*, 3.03% for *RIP5-OE*, 29.29% for *rel1-D*, and 27.78% for *rel1-D/rip5-ko* (Figure 7A,B). Compared with ZH11, the survival rates of *rip5-ko*, *rel1-D* and *rel1-D/rip5-ko* were all improved, and there was no significant difference, whereas *RIP5-OE* showed more severe wilt and lower survival than ZH11 (Figure 7A,B). The above results suggest that *REL1* and *RIP5* regulate drought tolerance through the same pathway, and that *REL1* may regulate drought tolerance by regulating the function of *RIP5*.

To better determine the possible functions of REL1 and RIP5 during the drought-stress response, we further measured AsA content and AAO activity at different times in ZH11, *rip5-ko*, *RIP5*-OE, and *rel1-D/rip5-ko* lines under drought-stress treatment. Generally, AAO activity increased in the first 3 h after treatment, AsA content also increased 3 h after treatment, and the *RIP5-OE* line displayed a higher AAO activity increase than the WT, *rip5-ko*, and *rel1-D/rip5-ko* lines (Figure 7C,D). After 6 h of drought-stress treatment, AAO activity was highest in the *RIP5-OE* line, followed by ZH11 and then *rip5-ko*, and the *rel1-D/rip5-ko* line showed the lowest AAO activity. In contrast, AsA content showed exactly the opposite results, with the highest AsA content measured in the *rel1-D/rip5-ko* line and the lowest in the *RIP5-OE* line. The AAO activity in each sample decreased significantly at 12 h (Figure 7C). Compared with control, the AsA content of ZH11, *rip5-ko*, *RIP5-OE*, *rel1-D*, and *rel1-D/rip5-ko* lines were all significantly increased. In contrast to AAO activity, the AsA content of all samples decreased between 12 h and 24 h (Figure 7D). These results further imply that *REL1* is involved in regulating drought tolerance by inhibiting *RIP5*.

### 3.7. Analysis of Transcriptome Assay between rip5-ko and ZH11 under PEG Stress

To further investigate the underlying mechanism of drought tolerance regulated by *RIP5*, the ZH11 and *rip5-ko* lines were treated with PEG at seedling stages and then subjected to RNA-sequencing. A total of six libraries were constructed, and 39.95 GB of clean reads were obtained (Appendix A). Then, DEGs were determined with criteria of padj < 0.05 and |log2 (fold change)| ≥ 1 set as thresholds for transcriptome analysis. As a result, 260 DEGs (158 up-regulated and 102 down-regulated) were identified between ZH11_PEG and *rip5-ko*_PEG (Figure 8A) samples, which might improve the growth state of the whole mutant plant. These DEGs were related to different KEGG pathways that were mainly involved in metabolism and environmental information processing, such as the biosynthesis of secondary metabolites; the biosynthesis of various plant secondary metabolites; metabolic pathways; flavonoid biosynthesis; the citrate cycle (TCA cycle); galactose metabolism; starch and sucrose metabolism; the MAPK signaling pathway; plant amino sugar and nucleotide sugar metabolism; alanine, aspartate and glutamate metabolism; and tryptophan metabolism (Figure 8B). These pathways are all related to ascorbate metabolism. Gene Ontology (GO) enrichment analysis of DEGs between ZH11_PEG and *rip5-ko*_PEG samples showed different functional terms related to abiotic stress and antioxidants, such as response to stimulus, detoxification, antioxidant activity, and catalytic activity (Figure 8C), prompting us to speculate that the *rip5-ko* mutation might affect the content of ascorbate to enhance drought tolerance.

We tested the expressions of several drought-induced genes from DEGs in these plants under PEG treatment, including *OsWRKY76*, *OsbHLH148*, *POX22.3*, *OsMYB2*, *OsLEA3* and *OsDT11.* The relative expression of *OsWRKY76* in *rip5-ko* samples is about twelve times that in ZH11 samples, whereas *OsbHLH148* is ten times, *POX22.3* is three times, *OsMYB2* is two times, *OsLEA3* is a hundred times, and *OsDT11* is ten times that in ZH11 samples. The relative expression levels of all genes in *rip5-ko* samples were significantly higher than those in ZH11 samples under PEG treatment (Figure 9A–F). The above results indicate that these genes could be possibly influenced by *RIP5*, and that enhanced drought tolerance of the *rip5-ko* mutants leads to substantially higher endogenous expression of drought-responsive genes, eventually resulting in the tolerance of *rip5-ko* to drought.

## 4. Discussion

Rice *(Oryza sativa* L.) is one of the most important food crops worldwide. In comparison to other crops, rice is more susceptible to drought and needs a lot of water during planting. Drought has an impact on rice’s morphological, physiological, biochemical, and molecular characteristics as well as its quality and production characteristics. Crops have evolved sensitive systems to withstand the effects of drought. Clarifying the specific mechanisms underlying the drought response of crops will speed up the development of drought-resistant cultivars and offer valuable insights for molecular breeding [21,22,23,24]. In addition, due to fast-paced industrialization and urbanization changes as well as escalating climatic changes, agricultural water is becoming increasingly scarce [25]. Therefore, it is very important to elucidate relevant mechanisms for improving drought tolerance in rice. In this study, we present multiple lines of evidence supporting our finding that *RIP5* negatively regulates drought-stress tolerance. Firstly, the expression of *RIP5* was induced by drought stress, suggesting that RIP5 is a drought-response factor in rice (Figure 2C). Secondly, the activity of AAO decreased and the content of AsA increased in *rip5-ko* plants after PEG treatment (Figure 6). Moreover, *rip5-ko* mutants displayed significantly increased survival rates compared to WT plants (Figure 4C,D). Last but not least, the activity of AAO increased and the content of AsA decreased in *RIP5-OE* plants after PEG treatment (Figure 5). However, *RIP5-OE* mutants displayed significantly decreased survival rates compared to WT plants (Figure 5A,C). Finally, in RNA-sequencing most of the DEGs are related to stress responses, and *rip5-ko* might affect the content of ascorbate to enhance drought tolerance (Figure 8A–C). Our results also demonstrated that *OsWRKY76*, *OsbHLH148*, *POX22.3*, and *OsMYB2* genes, functioning as positive regulators in the drought response [26,27,28,29], were significantly up-regulated in *rip5-ko* plants (Figure 9A–D). Taken together, these results indicate that *RIP5* negatively regulates drought-stress tolerance in rice.

It was revealed that *RIP5* encodes AAO and the AAO protein belongs to the multi-copper oxidase family of homodimeric enzymes. The enzyme was generally present in the apoplastic space of plant cells [9]. The connection between AAO activity and developmental processes, the apoplastic redox state, and particularly stress tolerance, was established in plants [15,16]. AsA was oxidized to dehydroascorbate (MDHA) via AAO [30]. AsA is a universal non-enzymatic antioxidant molecule that serves as an essential substrate for the detoxification of reactive oxygen species [31]. AsA plays crucial roles in scavenging ROS and it also extensively participates in plant growth, development, signal transduction, and stress tolerance [32,33,34]. Exogenously applied AsA protects proteins and lipids against salinity- and drought-induced oxidative stress [31,35]. The *rel1-D* line has a strong resistance to drought when compared to the ZH11 line, and it exhibits markedly increased expression of ABA-related genes to promote the drought response [15,16,17]. In this study, we revealed that REL1 interacts with RIP5 in the chloroplasts. Bioinformatics analyses indicated that RIP5 encodes AAO (Appendix A), which consists of three Cu-oxidase domains (Appendix A), suggesting that RIP5 may have functions in response to abiotic stress. Also, RIP5 belongs to the OsAAOs family, and the expression of *OsAAOs* in drought-stress tolerance suggests that they may play roles at different times or in different degrees of drought treatment (Figure 2B–G).

One consequence of drought stress was enhanced ROS production in different cellular compartments, such as peroxisomes, chloroplasts, and mitochondria. ROS scavenging and detoxification are achieved by non-enzymatic antioxidants comprising AsA, GSH, flavonoids, and carotenoids [12]. Numerous studies have established a direct correlation between the level of induction of the antioxidant system and the degree of drought tolerance in plants [36]. In our study, we found that *RIP5* negatively regulates drought tolerance. In addition, *REL1* positively regulates drought tolerance [35,37]. Given that non-enzymatic antioxidants are extensively involved in drought response in plants, we also measured AAO activity and AsA content in ZH11, *RIP5*-OE, *rip5-ko* and *rel1-D/rip5-ko* lines, and the results showed that the survival rates of *rip5-ko*, *rel1-D*, and *rel1-D/rip5-ko* plants were all improved compared to ZH11 plants. Both AAO activity and AsA content increased in the first 3 h after treatment, and the *RIP5-OE* line displayed the highest AAO activity increase compared to the ZH11, *rip5-ko*, and *rel1-D/rip5-ko* lines (Figure 7C,D). In contrast to AAO activity, the AsA content of all materials decreased between 12 h and 24 h (Figure 7D). These results imply that *REL1* is involved in regulating drought tolerance by inhibiting *RIP5*.

Collectively, a proposed model was given to explain the roles of the REL1 and RIP5 module in regulating drought tolerance in rice. Drought stress leads to osmosis stress, and osmosis stress intensifies water deficiency and cause the excessive accumulation of ROS that eventually disrupt the redox homeostasis essential for cell metabolism. In our study, under drought-stress conditions, rice induced *REL1* expression and inhibited the expression of *RIP5*, and the decreased expression of *RIP5* increased the content of AsA. In vivo, the level of AsA content of may affect ROS clearance and thus biological resistance, and may ultimately weaken the drought response (Figure 10).

## Figures and Tables

**Figure 1 cells-13-00887-f001:**
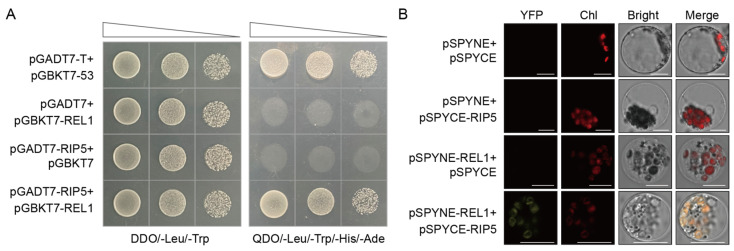
REL1 physically interacted with RIP5. (**A**) REL1 interacted with RIP5 in Y2H assay. Yeast transformants were cultured on DDO/-Leu/-Trp and QDO/-Leu/-Trp/-His/-Ade selection media. pGADT7-T/pGBKT7-53 was used as a positive control, and both pGADT7/pGBKT7-REL1 and pGADT7-RIP5/pGBKT7 were used as negative controls. (**B**) BiFC assays showed the interaction between REL1 and RIP5 in rice protoplasts. The combinations of pSPYNE with pSPYCE, pSPYNE with pSPYCE-RIP5, and pSPYNE-REL1 with pSPYCE were used as negative controls. Chl—Chlorophyll; Scale bar, 10 μm.

**Figure 2 cells-13-00887-f002:**
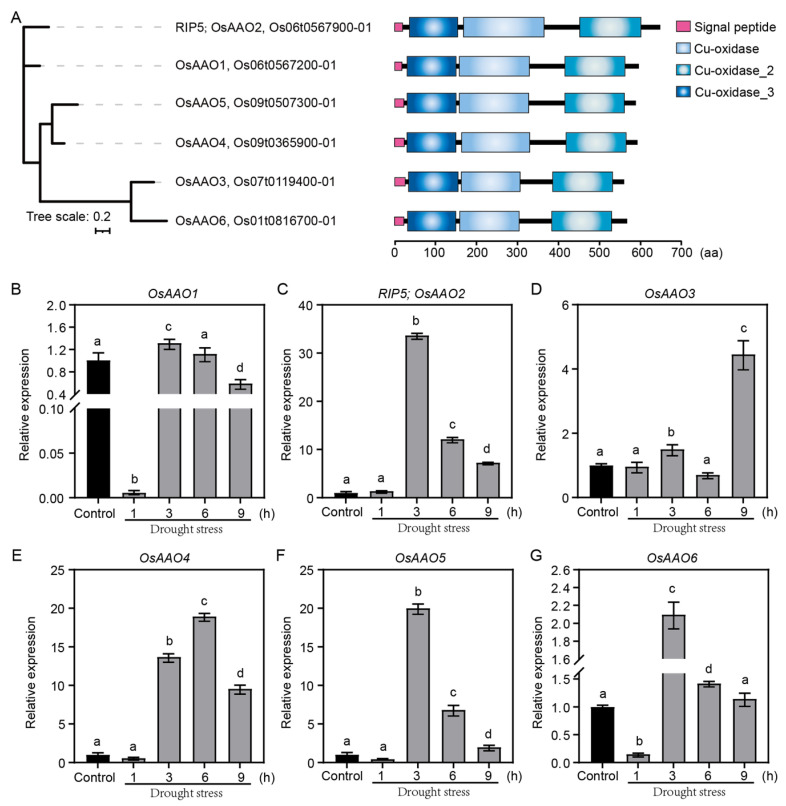
Phylogenetic tree, protein domain structure, and expression patterns of the OsAAOs family. (**A**) Phylogenetic tree and the protein domain structures of the OsAAOs family, tree scale, 0.2. (**B**–**G**) Relative expression of the *OsAAOs* family under control conditions and a time course of drought-stress treatment in rice, LSD and Duncan, “a, b, c, d” indicates that *p* < 0.01, the difference is extremely significant.

**Figure 3 cells-13-00887-f003:**
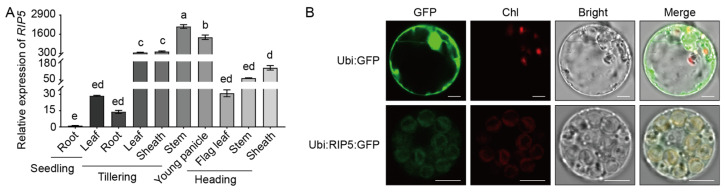
Tissue-specific expression pattern analysis and subcellular localization of RIP5. (**A**) Tissue-specific expression pattern of *RIP5* in different developmental stages and organs of ZH11. LSD and Duncan, “a, b, c, d, e” indicates that *p* < 0.01, the difference is extremely significant. (**B**) Subcellular localization of RIP5 in rice protoplasts cells. Scale bar, 10 μm.

**Figure 4 cells-13-00887-f004:**
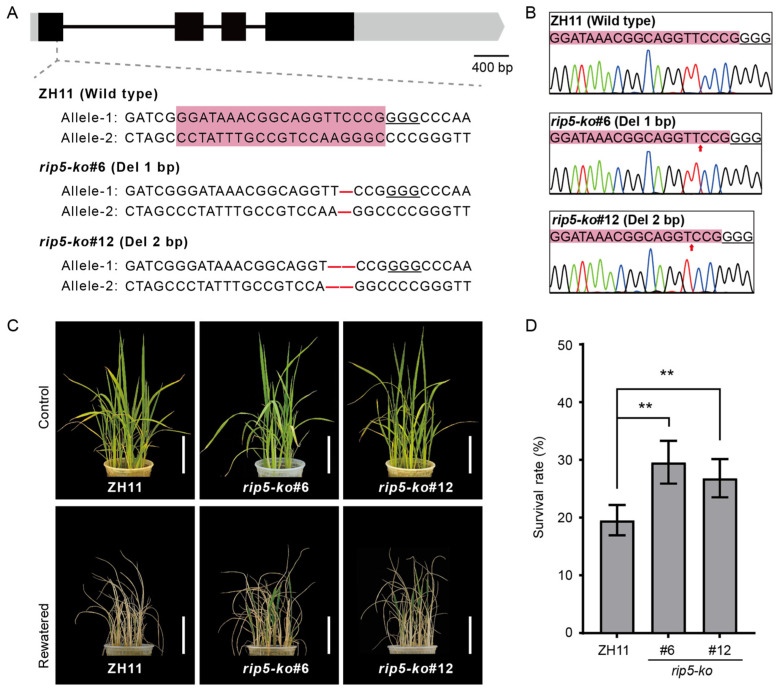
CRISPR/Cas9-mediated targeted mutagenesis of *RIP5* and drought resistance of *rip5-ko* compared to ZH11. (**A**) CRISPR/Cas9-mediated targeted mutagenesis of *RIP5.* The schematic diagram shows the *RIP5* gene with the CRISPR/Cas9 target site indicated by the dotted line. Exons and introns are represented by black boxes and lines. Each alignment between *rip5-ko* lines and ZH11 sequences containing the target sites is shown below the schematic diagrams. Sketches of CRISPR/Cas9 target sites of *RIP5* and the sgRNA target site is in pink highlights. The PAM motif is underlined. Scale bar, 400 bp. (**B**) Two types of *rip5-ko* rice mutants with 1 bp or 2 bp deletion mutations in CRIPSPR/Cas9 were confirmed using sequencing. The PAM motif is underlined. The sgRNA target site of ZH11 and two *rip5-ko* rice mutants is in pink highlights. The red arrow shows where the base is missing. (**C**) Drought-stress treatment in WT and *rip5-ko*. Upper: one-month-old plants before drought. Lower: drought for 14 d then recovery for 7 d. Scale bar, 10 cm. (**D**) Survival rates after 14 d of drought-stress treatment and 7 d of recovery. Values are shown as mean ± SD, *n* = 30, *t*-test, and “**” indicates that *p* < 0.01, the difference is extremely significant.

**Figure 5 cells-13-00887-f005:**
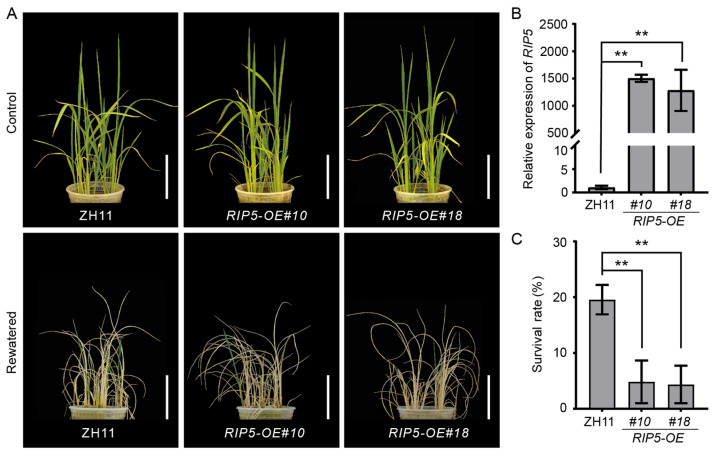
Overexpression of *RIP5* decreases drought-stress tolerance in rice. (**A**) Drought-stress treatment in ZH11 and *RIP5-OE*. Upper: one-month-old plants before drought. Lower: drought for 14 d then recovery for 7 d. Scale bar, 10 cm. (**B**) Expression level of *RIP5* in different *RIP5-OE* lines was detected by qRT-PCR. (**C**) Survival rates of ZH11 and *RIP5-OE* after 14 d of drought-stress treatment and 7 d of recovery. Values are shown as mean ± SD, *n* = 30, *t*-test, and “**” indicates that *p* < 0.01, the difference is extremely significant.

**Figure 6 cells-13-00887-f006:**
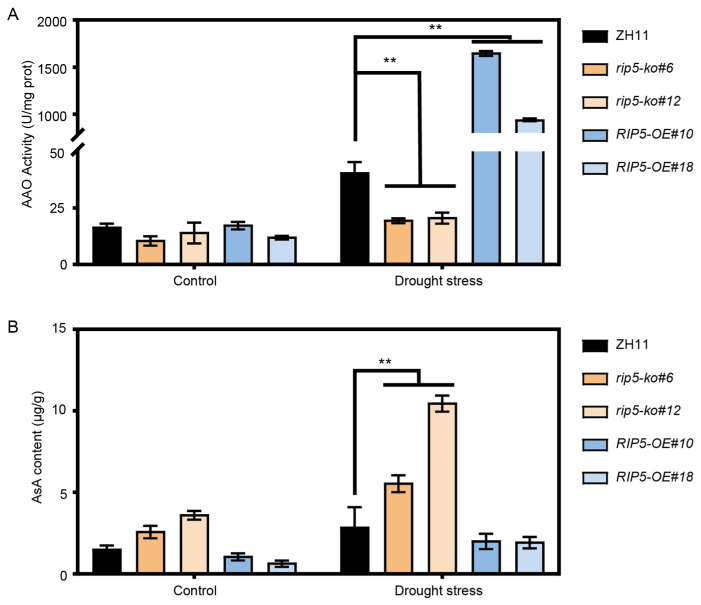
The AAO activity and AsA content of ZH11, *rip5-ko* and *RIP5-OE* lines under drought-stress treatment. (**A**) The AAO activity in ZH11, *rip5-ko* and *RIP5-OE* lines under drought-stress treatment. (**B**) The AsA content in ZH11, *rip5-ko* and *RIP5-OE* lines under drought-stress treatment. Values are shown as mean ± SD, *n* = 20, *t*-test, and “**” indicates that *p* < 0.01, the difference is extremely significant.

**Figure 7 cells-13-00887-f007:**
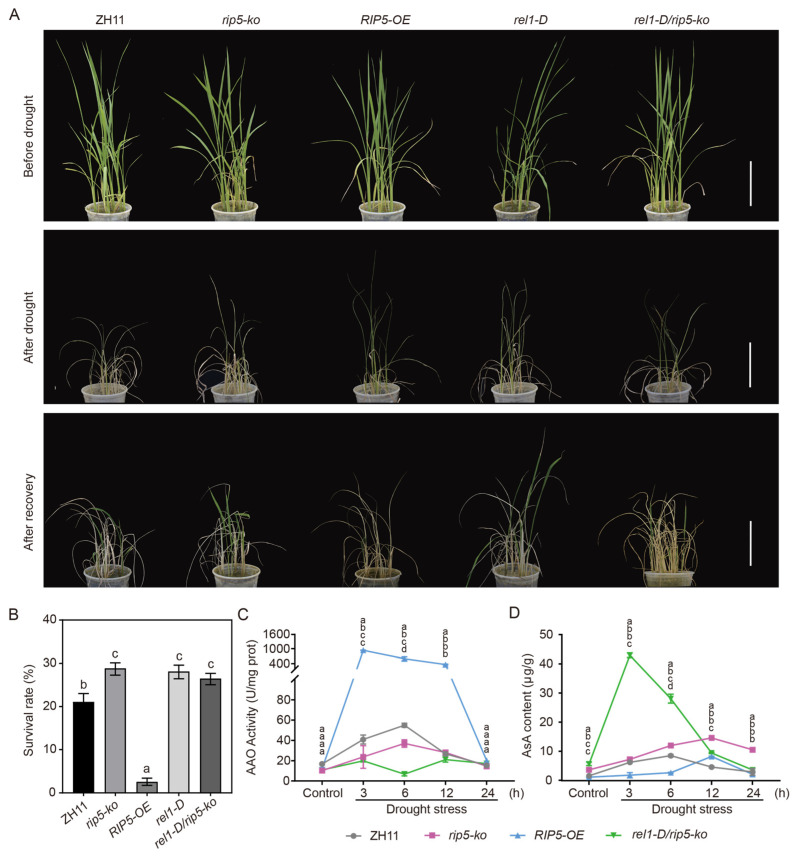
*REL1* is involved in regulating drought tolerance by inhibiting *RIP5*. (**A**) Drought-treated plants of ZH11, *rip5-ko*, *RIP5*-OE, *rel1-D*, and *rel1-D/rip5-ko* rice lines. Top row: one-month-old plants before drought treatment. Middle row: after drought treatment for 14 d. Lower row: after drought treatment for 14 d then recovery for 7 d. Scale bar, 10 cm. (**B**) Survival rates of ZH11, *rip5-ko*, *RIP5*-OE, *rel1-D,* and *rel1-D/rip5-ko* lines after 14 d of drought-stress treatment and 7 d of recovery. (**C**) AAO activity and (**D**) AsA content at different times during drought-stress treatment of ZH11, *rip5-ko*, *RIP5*-OE, *rel1-D* and *rel1-D/rip5-ko* lines. LSD and Duncan, “a, b, c, d” indicates that *p* < 0.01, the difference is extremely significant.

**Figure 8 cells-13-00887-f008:**
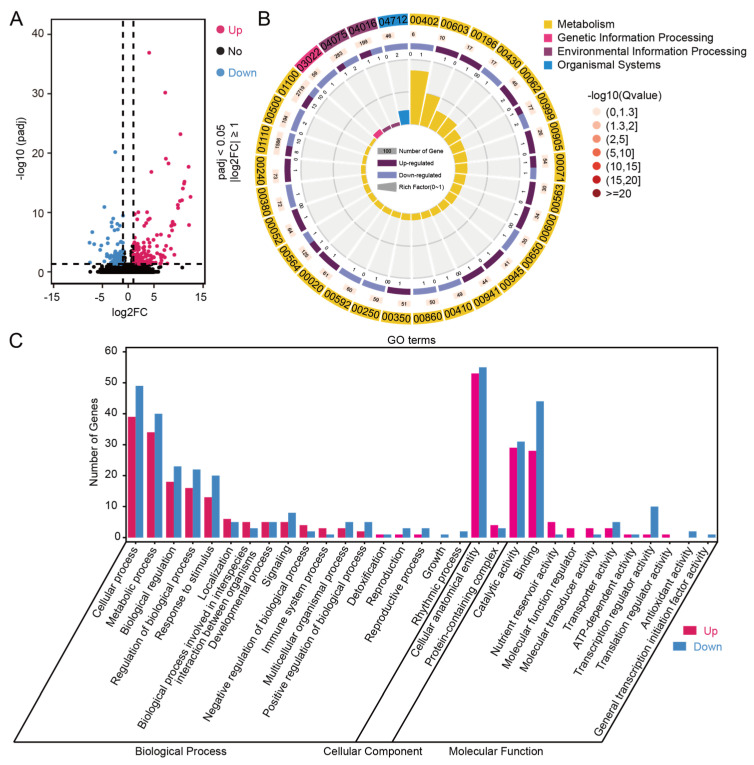
Comparison of transcriptome analysis results between *rip5-ko* and ZH11 plants under PEG stress. (**A**) Volcano plot of DEGs that were differentially expressed between ZH11_PEG and *rip5-ko*-PEG. The red dots on the right of the plot represent significantly up-regulated genes, the blue dots on the left of the plot represent significantly down-regulated genes, and the black dots in the middle of the plot represent genes that did not show significant differential expression. (**B**) KEGG metabolic pathway enrichment analysis comparing DEGs between ZH11_PEG and *rip5-ko*_PEG samples. (**C**) GO functional classification enrichment analysis comparing DEGs between ZH11_PEG and *rip5-ko*_PEG samples.

**Figure 9 cells-13-00887-f009:**
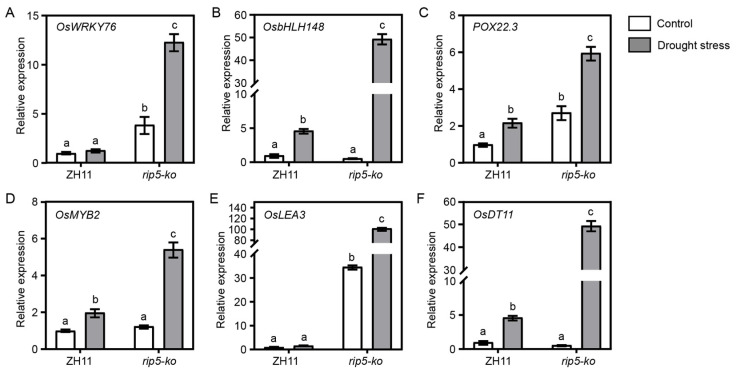
Expression level of genes in ZH11 and *rip5-ko* lines after PEG treatment. Relative expression level of (**A**) *OsWRKY76*, (**B**) *OsbHLH148*, (**C**) *POX22.3*, (**D**) *OsMYB2*, (**E**) *OsLEA3*, and (**F**) *OsDT11* genes in ZH11 and *rip5-ko* lines after PEG treatment. LSD and Duncan, “a, b, c” indicates that *p* < 0.01, the difference is extremely significant.

**Figure 10 cells-13-00887-f010:**
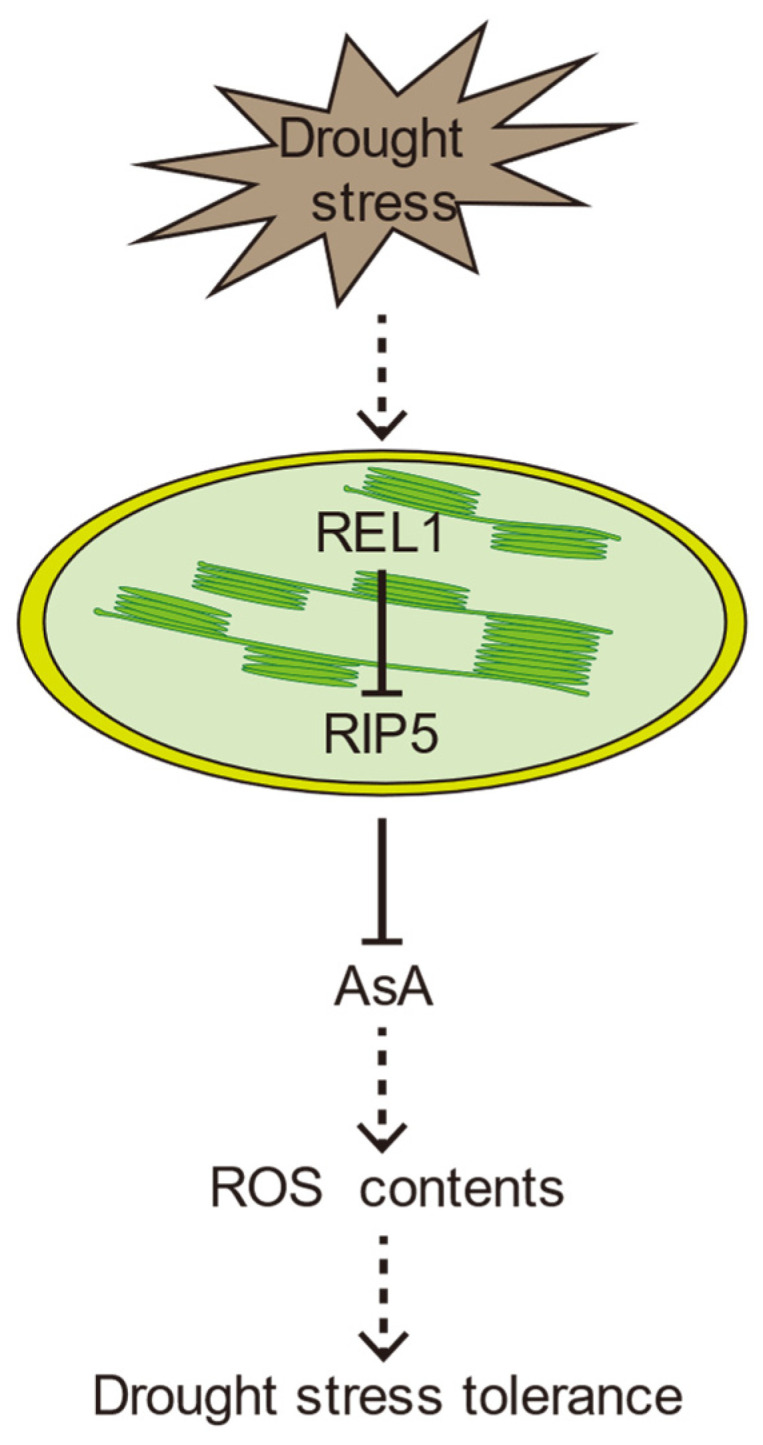
A hypothesized working model depicting the roles of the *REL1* and *RIP5* module in regulating drought tolerance in rice.

## Data Availability

All RNA-sequencing data are available from the Sequence Read Archive (SRA) data repository (accession PRJNA745033) of the NCBI.

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
