# Peer review of "RIP5 Interacts with REL1 and Negatively Regulates Drought Tolerance in Rice"

_cells, 2024, doi:10.3390/cells13110887_

Round 1
Reviewer 1 Report (Previous Reviewer 1)
Comments and Suggestions for Authors
The manuscript “RIP5 Interacts with REL1 and Negatively Regulates Drought Tolerance in Rice” by Zhang et al. presents an investigation into the possibility of RIP5 and REL1 interacting for the further the regulation of signaling pathways in in rice drought response.
First of all, in the Introduction section the data of which is known about RIP5 from literature is completely absent. This makes it difficult to appreciate what the authors repeatedly claim that RIP5 is ascorbate oxidase. Are there any data from literature, which confirm that? If there is no data in the literature confirming this, then data that the exons of ascorbate oxidases and RIP 5 are located in a similar way (Figure 2 A) and that the intensity of expression of the genes encoding them changes in a similar way (Figures 2C- is not enough to prove that RIP5 is AAO. The data of nucleotide sequences alignment of in the genes encoding RIP5 is AAO should be given. In addition, it would be desirable to demonstrate AAO activity of the RIP5 protein. More convincing in this sense are the data in Figure 6A. Nevertheless, it can be assumed that a decrease in AAO activity in knockouts and an increase in overexpressants is a secondary consequence of the functioning of signaling pathways induced by RIP5 knockout and overexpression. Anyway, AAO activity of RIP5 protein is something that needs to be proven, and not stated as a proven fact from the very beginning of the presentation of the results.
The style of presentation of the Results is unacceptable. It is necessary to describe the results based on the data in the figure in accordance with their numbers. Whereas in this manuscript the authors immediately discuss the results, bypassing any description of them, starting from the Figure 2B, than Figure 4A, etc.
The names of the Figures are weird. “Figure 2. RIP5 is a member of OsAAOs family and expression patterns of OsAAOs family”. It does not make sense. Except, the Figure name should not have any conclusions. It should describe the data of the Figure itself.
“Figure 3. Expression pattern analysis of RIP5”. Figure 3B is not the expression pattern.
“Figure 4. Drought resistance of rip5-ko compared to ZH11.” Figures 4A, 4B are not the data of drought resistance of rip5-ko compared to ZH11. The data of these Figures should be “The Figure 1” and should have reference in L.105.
“Figure 6. Analysis of antioxidant enzyme activities”. As far as we can see, here is shown the data of AAO which is the oxidant enzyme, whereas AsA is antioxidant indeed, but not an enzyme.
I also don’t understand the data of the Figure 1B. L. 382: “YC/YN, YN/RIP5-YC, REL1-YN/YC were used as negative control”. What does it mean? How do the data of Figure 1B REL1-YN/YC correspond with the data of Figure 3B? L. 472-473: “The enzyme was generally present in the apoplastic space of plant cells [8].” At that, RIP5 is located in chloroplasts, according to your data. What do you think about that? It is unusual for AAO? Why do all AAOs genes possess the signal peptides in this case? (Figure 2B).
All these comments mean the section Results needs to be completely rewritten, because it is not the Discussion. For the moment, the section Results is absent. By the way, L. 209: “3.1. Subsection”. What does it mean?
L. 16: ROLLED AND ERECT LEAF1 (REL1)”. The word “protein” is missed.
L. 19: “REL1 interacting protein 5 (RIP5), which encodes ascorbate oxidase (AAO)”. Proteins are not able to encode other proteins.
L. 20: We found that RIP5 was strongly expressed under drought stress conditions” in the WT plants?
L. 20: “rip5-ko significantly improved the tolerance of rice plants to drought”. rip5-ko is the name of the mutants. Did you mean the knockout of the RIP5 encoding gene has led to the improvement of the tolerance of rice plants to drought?
L. 23: “RIP5 negatively regulated drought tolerance in rice by enhancing AAO activity” Regarding that RIP5 is probably AAO, did you mean that RIP5 enhanced itself AAO activity, or did it enhance the activity of other AAOs? Which data show that?
L. 24: “scavenging ascorbic acid (AsA) thereby reducing ROS clearance”. Rephrase. The expression is unclear. Usually the word “scavenging” used to describe the process of reactive oxygen species quenching by antioxidants, but it is unclear, how ascorbate can be quenched. What did you mean speaking “reducing ROS clearance”, is unclear.
L. 27-28: “reveal that revealing”
L.47-51. The references should be added. Besides, the statement that “plants have evolved ROS scavenging non-enzymatic systems to maintain the homeostasis of ROS, which mainly include ascorbic acid (AsA), glutathione (GSH), mannitol and flavonoids” is not precise. The review by Rudenko et al. Antioxidants of Non-Enzymatic Nature: Their Function in Higher Plant Cells and the Ways of Boosting Their Biosynthesis. Antioxidants (Basel). 2023 Nov 17;12(11):2014. doi: 10.3390/antiox12112014. can help.
L. 62. The reaction, catalyzed by AAOs should be added.
L. 76. The paragraph, concerning the known from literature information about RIP5 should be added.
L. 77: Previously, we previously identified
L. 79: “with significantly increased expression” In what conditions?
L. 83-84. “We constructed three types of transgenic plants, including rip5-ko mutant lines, RIP5-OE and rel1-D/rip5-ko”. But you have also used the rel1-D mutants - L. 90: “Rice plants including ZH11, rip5-ko, RIP5-OE, rel1-D and rel1-D/rip5-ko used in this study..” What are these mutants? What gene was knocked out? Was it REL1 protein encoding gene? This should be stated on first mention
L. 97. “Transgene constructs” The mutants obtained using the Cripsr/Cas9 gene editing method are not usually called transgenes. Except, you describe rather the obtainment of the mutants, not the constructs.
L. 98-111: How rel1-D mutants were obtained?
L. 103-104: “using an agrobacterium-mediated method by Wuhan Boyuan Company” What does it mean?
L. 119-121. The reference is missed.
L. 123 Rice protoplasts were extracted and transformed as described previously [16]. The reference [16] have no any information about protoplasts. L. 145-146: Protoplasts were isolated using a protocol modified by Chen et al [18]. Where protoplasts isolated in different ways for Subcellular localization of RIP5 and for Bimolecular fluorescence complementation?
L. 381: “BiFC assays show that the interaction between REL1 and RIP5 in rice protoplasts.” What does it mean? The sentence seems incomplete.
L. 461. “However..” What is the contradiction here?
Comments on the Quality of English LanguageThere are awkward English expressions, I have listed in the main part of review.
Author Response
We would sincerely thank the reviewer for the time and effort that you have put into reviewing the previous version of this manuscript. Your suggestions have enabled us to improve our work. Accordingly, we have carefully revised and proofread the manuscript, and the detailed corrections are listed below which we hope to meet with approval. Please see the attachment.

Reviewer 2 Report (Previous Reviewer 2)
Comments and Suggestions for Authors
Zhang et al.: RIP5 interacts with REL1 and negatively regulates drought tolerance in rice
The RIP5 gene's interaction with REL1 and its negative regulation of drought tolerance in rice is an intriguing aspect of plant biology. Such interactions between genes and regulatory pathways play a significant role in determining a plant's response to environmental stresses like drought. Understanding these mechanisms could potentially lead to the development of more resilient crop varieties through genetic engineering or breeding programs. It highlights the complexity of plant stress responses and the importance of unraveling these molecular networks for agricultural improvement.
The manuscript is well-written. A very large amount of work was involved in the study and the manuscript contains valuable results. It means that the authors have taken great care in designing experiments, collecting data, and analyzing results to ensure the reliability and validity of their findings. Well-designed research forms the foundation for further advancements and applications in the field, such as developing drought-tolerant rice varieties to address global food security challenges.
However, the manuscript needs some minor improvements before publication.
Please check the Oxford comma throughout the manuscript.
Introduction:
Line 77: Check the sentence, the authors used twice “previously”.
Materials and Methods:
Line 181, 188, etc.: Please do not start a sentence with an abbreviation, check the whole manuscript.
Results:
Please rearrange the tables and figures. e.g. Figure 2B and Figure 4 appear earlier than Figure 1.
Line 305, 314, 364, and 366: Please do not use any citations in the Results section. Present only the research data. The authors can move these sentences to the Discussion section.
Author Response
We thank the reviewer for the time and effort that you have put into reviewing the previous version of the manuscript. Your suggestions have enabled us to improve our work. Please see the attachment.

Round 2
Reviewer 1 Report (Previous Reviewer 1)
Comments and Suggestions for Authors
Dear Authors!
You have answered all my comments.
I have the only one suggestion. Please add to the Supplementary both Figures that you attached to the response to my comments. This is the important information for a reader for understanding the results of the present study.
Author Response
Thank you for your suggestion. Please see the attachment.

This manuscript is a resubmission of an earlier submission. The following is a list of the peer review reports and author responses from that submission.
Round 1
Reviewer 1 Report
Comments and Suggestions for Authors
Please see the attached document.

Comments on the Quality of English LanguageSome awkward expressions appeared in the text. It makes no sense to present them here, since the manuscript as a whole needs a thorough revision.
Reviewer 2 Report
Comments and Suggestions for Authors
Review on „REL1 interacting protein 5 (RIP5) encoding ascorbate oxidase mediates drought response in rice”.
Generally, I think that the topic of the paper is very interesting and has great importance. Overall the manuscript is clear, organized, and well-structured. The introduction was well-written however some more information needs to be added to it. Results are statistically analyzed but the accurate number of repetitions of each measured parameter is missing. The manuscript contains many valuable results. However, the manuscript needs some improvements and corrections.
Please check the Oxford comma throughout the manuscript. Please also check the ‘and’-es in the manuscript.
When the abbreviations are first used please write the full name of them out.
Specific comments:
Abstract:
Line 15: Why? What was the hypothesis? Why did the authors complete this work? Please add the aim and/or hypothesis of the work.
Line 17: „OsRIP5 may be involved in early plant growth and metabolism of AsA.” Please do not write „may be” in a scientific article. Please rewrite the sentence.
Keywords:
Please arrange the keywords in alphabetical order.
Introduction:
Line 28: Please add information about China's rice production (growing area, yield/ha, yield/year, etc.).
Line 38: Correct =>… reactive oxygen species (ROS)… Please use the full name of each abbreviation when used.
Line 44: Correct => … deoxyribonucleic acid (DNA)…
Line 59: Please do not start a sentence with an abbreviation, check the whole manuscript.
Please add the hypothesis and/or aim of the study.
Materials and Methods:
Please use subsections.
Line 133: Did 6% TCA base on w/v? Please add the information.
Line 143: Same comment as in Line 133.
Line 144: Please correct =>… including 3 h, 6 h, 12 h, and 24 h.
Line 161: Please use subscript for H2O2.
The presentation of statistical analysis is missing. Please add this information to the Materials and Methods section.
Results:
Line 176. Please delete. You do not need to indicate „subsection”.
Line 177: Please correct and use: 3.1. OsRIP5 interacts with OsREL1 and is localized in the chloroplast and correct the other subsections as well.
Line 179, 181: Please do not use literature references in the Results section only present the research data of this study.
Line 347: Please delete and insert figures and tables into their suitable places in the manuscript.
Please delete the supplementary materials from the manuscript these documents are already submitted separately as supplementary files.
Figures 1, 2, 3, and 4: Figures are very small and can’t see all data very well. Please make them bigger.
Figure 5B. Impossible to construe.
Figure 7: Too small figures.
Figure S1, and S2: How were measure the presented parameters? These plant characteristics are not shown in the Materials and Methods section. When (which phonological phase) and how these parameters were measured? What kind of equipments were used? like: ruler, analytical scale, etc.
Line 492. “More than 10 samples were used for statistical analysis.” This is not accurate. How many samples were used?
Line 480-486. Please put these in a Conclusion section and add some more conclusions as well.
Unfortunately, my main concern about this manuscript is that the clear and well-defined aim and hypothesis of this study are missing. In addition, the Materials and Methods section also needs improvement. Overall, I can suggest publishing this paper after a major revision.